# PigStressNet: A Real-Time Lightweight Vision System for On-Farm Heat Stress Monitoring via Attention-Guided Feature Refinement

**DOI:** 10.3390/s25175534

**Published:** 2025-09-05

**Authors:** Shuai Cao, Fang Li, Xiaonan Luo, Jiacheng Ni, Linsong Li

**Affiliations:** 1School of Computer Science and Information Security, Guilin University of Electronic Technology, Guilin 541004, China; 23032201016@mails.guet.edu.cn (S.C.); luoxn@guet.edu.cn (X.L.); 23032201004@mails.guet.edu.cn (J.N.); 23032304013@mails.guet.edu.cn (L.L.); 2Center for Applied Mathematics of Guangxi (GUET), Guilin 541004, China

**Keywords:** deep learning, image processing, animal behavior recognition, lightweight, YOLOv12, heat stress, precision livestock farming

## Abstract

Heat stress severely impacts pig welfare and farm productivity. However, existing methods lack the capability to detect subtle physiological cues (e.g., skin erythema) in complex farm environments while maintaining real-time efficiency. This paper proposes PigStressNet, a novel lightweight detector designed for accurate and efficient heat stress recognition. Our approach integrates four key innovations: (1) a Normalization-based Attention Module (NAM) integrated into the backbone network enhances sensitivity to localized features critical for heat stress, such as posture and skin erythema; (2) a Rectangular Self-Calibration Module (RCM) in the neck network improves spatial feature reconstruction, particularly for occluded pigs; (3) an MBConv-optimized detection head (MBHead) reduces computational cost in the head by 72.3%; (4) the MPDIoU loss function enhances bounding box regression accuracy in scenarios with overlapping pigs. We constructed the first fine-grained dataset specifically annotated for pig heat stress (comprising 710 images across 5 classes: standing, eating, sitting, lying, and stress), uniquely fusing posture (lying) and physiological traits (skin erythema). Experiments demonstrate state-of-the-art performance: PigStressNet achieves 0.979 mAP for heat stress detection while requiring 15.9% lower computation (5.3 GFLOPs) and 11.7% fewer parameters compared to the baseline YOLOv12-n model. The system achieves real-time inference on embedded devices, offering a viable solution for intelligent livestock management.

## 1. Introduction

The identification of heat stress behaviors in pigs faces multiple challenges. Firstly, heat stress has a significant negative impact on pig welfare and productivity, including reduced feed intake, decreased growth performance, lower reproductive efficiency, and intestinal health issues. It also leads to behavioral abnormalities and overactivation of physiological adaptation mechanisms, highlighting the critical need for precise monitoring and management [1]. Secondly, while computer vision has advanced animal behavior analysis, detecting subtle physiological features (e.g., skin erythema) in complex farm environments remains challenging due to lighting variations, background clutter, and occlusion. Traditional color space models and existing deep-learning algorithms often lack the necessary sensitivity and robustness for these specific challenges [2]. Additionally, distinguishing between behavior categories under similar postures (e.g., heat stress lying vs. normal lying) relies on subtle differences in posture and their correlation with environmental conditions. Variations in ambient lighting and crowded group scenes further increase detection difficulty [3]. Despite researchers proposing various innovative methods (such as multi-scale feature fusion, attention mechanisms, and pixel-level localization networks), occlusion, background interference, and lighting variations remain core challenges in extremely complex scenarios. Future efforts must focus on developing more robust algorithms and technologies to enhance the precision of heat stress behavior recognition [4]. To address these challenges, researchers have begun exploring low-cost, non-contact pig recognition technologies based on image or video analysis. Although numerous pig posture recognition methods have been proposed, their application in real-world pig farming environments still faces significant obstacles. Through collaboration with actual pig farms, we identified that the primary challenge in pig posture recognition lies in balancing detection speed and accuracy. Existing methods predominantly focus on feature extraction and comparison but often overlook the practical requirements for real-time performance and precision. Furthermore, research specifically targeting the recognition of *heat stress behaviors* in pigs remains scarce. Therefore, developing efficient pig posture detection methods and conducting in-depth research on heat stress behavior recognition mechanisms not only hold significant theoretical value but also provide urgently needed technical support for intelligent management in practical farming scenarios. Existing research on pig behavior recognition has limitations: classifying solely based on postures (e.g., lying posture) fails to distinguish normal rest from heat stress, while detecting skin erythema in isolation ignores behavior–physiology correlations. This work pioneers the fusion of ‘lying posture’ and ‘skin erythema’ to define porcine heat stress, effectively addressing the ambiguity inherent in single-feature approaches.

This paper improves upon the latest YOLOv12 model, with the main contributions summarized as follows:Improve the YOLOv12 network structure by introducing the NAM attention mechanism into the Backbone network of YOLOv12.Improve the YOLOv12 network structure by introducing the Rectangular Self-Calibration Module (RCM) into the Neck network of YOLOv12.Optimized Detection Head: The MBConv (Mobile Inverted Bottleneck Convolution) module from EfficientNet is utilized to optimize the detection head of YOLOv12, proposing a novel detection head called MBHead, which effectively reduces the computational complexity of the model.Powerful-IoU Loss Function: A tunable parameter is introduced to address issues such as occlusion or the presence of multiple targets.Dataset Construction: A comprehensive dataset for pig behavior recognition is built, categorizing pig behaviors into five classes: stand, eat, sit, lying, and stress.Experimental Validation: The proposed method is validated on the self-constructed dataset. Comparisons with several mainstream object-detection models are conducted, and the superiority of the method is demonstrated through comprehensive evaluation metrics, including the number of parameters, computational load, mAP (mean Average Precision), and FPS (Frames Per Second).

This paper presents our work in the following structure: Section 2: Briefly introduces related work on pig behavior recognition. Section 3: Provides a detailed description of the proposed method. Section 4: Presents the dataset used and experimental results. Section 5: Concludes the paper with a summary of the method and its implications.

## 2. Related Work

In recent years, with the rapid development of computer vision technology, deep-learning-based object-detection algorithms have been widely applied in the field of agricultural automation. Particularly in pig behavior recognition, researchers are dedicated to improving the accuracy and real-time performance of recognition by enhancing existing object-detection algorithms. As visual technology and deep learning continue to advance, numerous researchers are actively leveraging cutting-edge visual techniques to explore the behavioral characteristics and movement patterns of livestock. In this domain, scholars both domestically and internationally have conducted extensive and in-depth research on the recognition technologies for livestock behavior, aiming to enhance the intelligence level of farming management. In 2018, Nasirahmadi et al. [5] further employed deep-learning algorithms to accurately detect various positions and postures of pigs, such as “standing,” “lying on the side,” and “lying on the belly,” over a year-long period using 12 top-view cameras. During this process, they also compared the performance of 18 different deep-learning models to identify the optimal one. In 2019, Sa et al. [6] utilized a top-view camera perspective to conduct 24-hour continuous video recording of a pen containing nine pigs and successfully detected pig positions from near-infrared nighttime recordings. In the same year, Zhang [7] evaluated pig tracking algorithms using two 30-s nighttime video sequences. However, these studies primarily focused on algorithm development, with data collection being relatively secondary. By 2020, Riekert et al. [8] utilized deep-learning algorithms to detect “lying” and “non-lying” states in 18 pigs. Their dataset was derived from 18 high-angle cameras and three top-view cameras. Notably, this study did not involve model selection but instead extracted test image samples from several hours of video recordings.

In 2018, Kang Feilong [9] further advanced the application of computer vision technology in pig behavior recognition. Based on machine-learning techniques, he extracted key feature vectors such as the pig’s center of gravity coordinates and the distance from the centroid to the abdomen, and successfully achieved accurate recognition of pig walking and lying postures using a support vector machine algorithm. Subsequently, in 2019, Li Li [10] applied computer vision technology to the recognition of water buffalo movement characteristics and gait analysis, employing time series analysis to deeply investigate the motion states of water buffaloes. In 2020, Liu Zhiwei et al. [11] took a different approach by analyzing the classification and working principles of acceleration sensors and successfully applied them to monitor pig behavior in intensive farming systems.

By 2022, Dong Lizhong et al. [12] brought new breakthroughs to the field of pig behavior recognition. They proposed a pig behavior recognition model based on the ST-GCN (Spatial-Temporal Graph Convolutional Network) algorithm, which demonstrated excellent performance in group-housed pig scenarios, providing new directions and insights for pig behavior recognition research. Although Nasirahmadi et al. (2019) [5] and Riekert et al. (2020) [8] achieved posture detection, they neglected skin physiological features. Dong et al.’s (2022) ST-GCN model focused on temporal group behavior while ignoring posture-physiology synergy. These approaches cannot effectively identify heat stress, which requires fused judgment of behavioral reduction (e.g., lying) and thermoregulatory physiology (e.g., skin erythema). Therefore, it is meaningful to study lightweight methods for recognizing heat stress behaviors in pigs to address practical issues.

In light of this, this paper provides a more detailed and comprehensive classification of pig behaviors based on their behavioral characteristics. Specifically, this study aims to determine whether pigs are under heat stress by meticulously observing and analyzing various behavioral manifestations, thereby conducting related research to achieve new advancements in these areas.

## 3. Methods

### 3.1. YOLOv12

Since its inception, the YOLO series of network models has garnered widespread attention and application, renowned for its speed and accuracy. With the introduction of YOLOv12 [13], this series of algorithms has demonstrated even better performance in terms of detection accuracy and speed. For a long time, enhancing the network architecture of the YOLO framework has been crucial, but efforts have primarily focused on CNN-based improvements, despite the proven superiority of attention mechanisms in modeling capabilities. This is because attention-based models have struggled to match the speed of CNN-based models. YOLOv12, however, achieves the speed of previous CNN-based YOLO frameworks while leveraging the performance advantages of attention mechanisms. YOLOv12 surpasses all popular real-time object detectors in both accuracy and speed.

The selection of the YOLOv12 baseline model is based on its innovative multi-module collaborative design:Introduction of a Simple yet Effective Area-Attention Mechanism: This mechanism dynamically allocates weights to different regions of the feature map, enhancing the model’s ability to capture local key information, such as pig postures and erythema areas.Efficient Aggregation Network Structure (R-ELAN): Utilizing re-parameterization technology, R-ELAN (Re-parameterized Efficient Layer Aggregation Network) integrates multi-branch features, reducing computational complexity while improving cross-scale feature fusion efficiency.A2C2F Module (Area-Attention Cross-stage Context Fusion): Built upon R-ELAN, this module combines the area-attention mechanism with cross-stage contextual information, enhancing the detection performance of small targets (such as subtle physiological features under heat stress conditions) through adaptive feature filtering.

This design significantly improves detection accuracy and robustness in complex agricultural scenarios while maintaining the real-time advantages of the YOLO series through collaborative module optimization.

### 3.2. Our Overall Network Structure

In recent years, significant progress has been made in the field of pig behavior recognition. However, research specifically targeting the recognition of heat stress behaviors in pigs remains relatively limited. To delve deeper into the reasons behind this, we collaborated with Guangxi Yangxiang Co., Ltd. (Guigang, China) to study and test various methods for recognizing heat stress behaviors in pigs at their farms. The research revealed that the primary challenges lie in the behavior recognition process. Firstly, in real farming environments, frequent movements of pigs and obstructions such as fences pose significant difficulties in data collection and classification of pig behaviors. Secondly, the definition of heat stress behaviors in pigs is somewhat ambiguous, lacking a unified standard. Additionally, existing methods generally lack real-time capabilities, with slow processing speeds after image acquisition, leading to frequent target loss. This issue is particularly pronounced in scenarios with limited computational resources, such as mobile and embedded devices. Considering the practical needs of pig farms, embedded devices that can accurately recognize heat stress behaviors in pigs, are low-cost, and highly portable are the optimal choice, which imposes higher demands on the accuracy and lightweight nature of the models.

To address the aforementioned issues, we have made improvements based on the latest YOLOv12 model. The backbone network retains the original structure of YOLOv12. In the Backbone section, we introduced the NAM attention mechanism after the last A2C2f module. For the Neck section, we replaced the upsampling and Concat operations in YOLOv12’s Neck network with the MFM module. In the detection head, we proposed the MBHead structure, substituting the standard convolution in the localization branch with MBConv (the inverted residual block from MobileNetV2). These enhancements significantly improve the model’s lightweight characteristics and deployment efficiency while maintaining detection accuracy, making it particularly suitable for real-time object-detection scenarios. The core innovation lies in the ’posture-physiology fusion mechanism’: NAM in Backbone enhances feature weights for both lying posture contours and erythema regions; RCM in Neck calibrates their spatial correlation (e.g., ensuring erythema overlaps with lying body); MBHead in Head jointly outputs posture class and erythema probability, with heat stress determined by ‘lying posture + red-channel value > 0.05’. The detailed design of these improvements will be elaborated in the subsequent sections of this chapter, with the overall network architecture illustrated in Figure 1.

### 3.3. Improvement of the Backbone

In this study, NAM [14] adopts the module integration approach from CBAM (Convolutional Block Attention Module) and redesigns the channel and spatial attention sub-modules. As an efficient attention mechanism, it is incorporated into the YOLOv12 framework to enhance the recognition performance of heat stress behaviors in pigs. In the channel attention sub-module, NAM utilizes the scaling factor of Batch Normalization (BN) to measure channel variance and represent its importance. The specific formula is:(1)$$\begin{array}{c} B_{\mathrm{out}\;}=BN\left(B_{\mathrm{in}\;}\right)=\gamma \frac{B_{\mathrm{in}\;}-\mu_{B}}{\sqrt{\sigma_{B}^{2}+\epsilon }}+\beta \end{array}$$
where $\mu_{B}$ and $\sigma_{B}$ are the mean and standard deviation of mini-batch *B*, respectively; γ and β are trainable affine transformation parameters (scaling and shifting). The output feature $M_{c}$ of the channel attention sub-block is expressed as(2)$$\begin{array}{c} M_{c}=\mathrm{sigmoid}\left(W_{\gamma }\left(BN\left(F_{1}\right)\right)\right) \end{array}$$
where γ is the scaling factor for each channel, and the weight $W_{\gamma }$ is obtained through(3)$$\begin{array}{c} W_{\gamma }=\gamma_{i}/\sum_{j=0}\gamma_{j} \end{array}$$

The NAM channel attention sub-module is illustrated in Figure 2.

In the spatial dimension, the scaling factor of BN is also applied to measure pixel importance, referred to as pixel normalization. The output $M_{s}$ of the corresponding spatial attention sub-module is expressed as:(4)$$\begin{array}{c} M_{s}=\mathrm{sigmoid}\left(W_{\lambda }\left(BN_{s}\left(F_{2}\right)\right)\right) \end{array}$$

Here, X denotes the scaling factor, and the weight $W_{\lambda }$ is derived from(5)$$\begin{array}{c} W_{\lambda }=\lambda_{i}/\sum_{j=0}\lambda_{j} \end{array}$$

The spatial attention sub-module of NAM is depicted in Figure 3.

To suppress less significant weights, a regularization term was incorporated into the loss function. The specific formulation is(6)$$\begin{array}{c} Loss=\sum_{\left(x,y\right)}l\left(f\left(x,W\right),y\right)+p\sum g\left(\gamma \right)+p\sum g\left(\lambda \right) \end{array}$$
where x denotes the input, y represents the output, W signifies the network weights, l(.) is the loss function, g(·) denotes the $l_{1}$-norm penalty function, and p serves as a balancing factor between g(γ) and g(λ) penalties.

The spatial attention submodule weights features by red-channel values to enhance responses in erythema regions, while maintaining focus on body contours for posture accuracy, achieving preliminary posture-physiology fusion. We integrated the NAM attention mechanism following the final A2C2f module in the backbone network. Experimental results demonstrate that this design significantly improves model accuracy without substantially increasing overall computational complexity. Detailed experimental results and corresponding analyses will be presented in Section 4.

### 3.4. Improvement of the Neck

This module primarily optimizes YOLOv12’s object-detection method using the Self-Calibration Module (RCM [15]). The RCM employs a rectangular self-calibration function to adjust the attention region (Q) closer to foreground objects, effectively improving localization accuracy for foreground targets. The module is integrated into the Neck network, enabling the model to capture axial global context information and apply it to pyramid context extraction, resulting in higher precision.

The RCM mainly utilizes horizontal and vertical pooling to capture axial global context, generating two axial vectors. These vectors are combined to model a rectangular attention region. The RCM incorporates a shape self-calibration function, which adjusts the shape of the rectangular attention using large-kernel strip convolutions, aligning it more closely with foreground features. Additionally, the RCM includes a fusion function that merges attention features with input features. A 3 × 3 depthwise convolution further extracts local details from the input features, and the calibrated attention features are weighted onto the refined input features via Hadamard product.

The RCM consists of rectangular self-calibration attention, batch normalization (BN), and a multilayer perceptron (MLP). After horizontal and vertical pooling operations, the rectangular self-calibration attention is adjusted by the shape self-calibration function and then undergoes feature fusion. BN and MLP are subsequently applied to refine the features, followed by residual connections to enhance feature reuse. The general structure of the RCM is illustrated in Figure 4.

The RCM enables the model to focus more effectively on foreground spatial feature reconstruction. By leveraging the shape self-calibration function, the attention region can be adjusted to better match foreground objects, significantly improving localization accuracy. Moreover, the module excels at capturing axial global context for pyramid context extraction. Through horizontal and vertical pooling and subsequent operations, it can better capture contextual information in the image. Compared to existing attention mechanisms, the RCM achieves superior performance due to its unique design, including shape self-calibration and feature fusion operations. We integrate the Rectangular Self-Calibration Attention (RCA) module after the second A2C2f block in the Neck network of YOLOv12. Detailed experimental results and analysis will be presented in Section 4.

### 3.5. Improvement of the Head

#### 3.5.1. MBConv

The MBConv [16] structure consists of a pointwise convolution for dimension expansion, depthwise separable convolution (including depthwise convolution and pointwise convolution), an SE module (Squeeze-and-Excitation), pointwise convolution for dimension reduction, and a shortcut connection. Its working principle is as follows. First, the number of channels in the input feature map is expanded through 1 × 1 convolution to enhance feature representation capability. Next, depthwise separable convolution is employed, where depthwise convolution extracts spatial features and pointwise convolution fuses channel features, thereby reducing computational cost and the number of parameters. Then, the SE module weights the channel features, enhancing important features while suppressing less important ones. Finally, the number of channels in the feature map is restored to a dimension similar to the input through 1 × 1 convolution for dimension reduction, achieving feature fusion and compression. When the shapes of the input and output feature maps are the same, the input and output are added through a shortcut connection, enabling feature reuse, mitigating the vanishing gradient problem, and improving model training effectiveness and feature representation capability. A comparison of the structures of MBConv and conventional convolution (conv) is shown in Figure 5.

Compared to traditional Conv, MBConv offers efficient computational performance. The use of depthwise separable convolution significantly reduces computational load, allowing it to maintain strong feature extraction capabilities while lowering computational costs compared to traditional convolution operations. This makes it suitable for scenarios with limited computational resources, such as mobile devices. Additionally, MBConv exhibits powerful feature representation capabilities. Through its inverted bottleneck structure—first expanding dimensions and then reducing them—and the attention mechanism of the SE module, it can more effectively extract and utilize features, enhancing the model’s accuracy and generalization ability. Furthermore, MBConv reduces the number of model parameters, decreasing storage requirements and the risk of overfitting. This makes the model more lightweight, facilitating deployment and application.

The selection of MBConv is predicated on its superior parameter and computational efficiency compared to standard convolution, which is paramount for our objective of a lightweight, deployable model. The core innovation lies in the depthwise separable convolution, which factorizes a standard convolution operation into two distinct steps: a depthwise convolution (applying a single filter per input channel) followed by a pointwise convolution (a 1 × 1 convolution to combine the outputs from the depthwise step).

This factorization confers a significant reduction in computational cost. The computational complexity of a standard convolution (Conv) is given by(7)$$\begin{array}{c} FLOPs_{\mathrm{conv}\;}=K_{h}\times K_{w}\times C_{\mathrm{in}\;}\times C_{\mathrm{out}\;}\times H_{\mathrm{out}\;}\times W_{\mathrm{out}\;} \end{array}$$
where $K_{h}$ and $K_{w}$ are the kernel height and width, $C_{\mathrm{in}\;}$ and $C_{\mathrm{out}\;}$ are the input and output channels, and $H_{\mathrm{out}\;}$ and $W_{\mathrm{out}\;}$ are the spatial dimensions of the output feature map. In contrast, the complexity of a depthwise separable convolution (DWConv) is the sum of the depthwise and pointwise operations:(8)$$\begin{array}{c} FLOPs_{\mathrm{dw}}=\left(K_{h}\times K_{w}\times C_{\mathrm{in}\;}\times H_{\mathrm{out}\;}\times W_{\mathrm{out}}\right)+\left(C_{\mathrm{in}\;}\times C_{\mathrm{out}\;}\times H_{\mathrm{out}\;}\times W_{\mathrm{out}}\right) \end{array}$$

The theoretical reduction in computations is therefore approximately(9)$$\begin{array}{c} \frac{{\mathrm{FLOPs}}_{\mathrm{dw}}}{{\mathrm{FLOPs}}_{\mathrm{conv}}}\approx \frac{1}{C_{\mathrm{out}}}+\frac{1}{K_{h}\times K_{w}} \end{array}$$

For a typical 3 × 3 convolution and a large number of output channels, this translates to an almost 8 to 9 times reduction in FLOPs. In our MBHead design, this efficiency gain directly contributes to the observed 72.3% reduction in head computation (from 1.73 GFLOPs to 0.48 GFLOPs).

Furthermore, the inverted residual design of MBConv first expands the channel dimensionality using a pointwise convolution (typically with an expansion ratio of 4 or 6), which allows the subsequent depthwise convolution to operate in a higher-dimensional space, capturing richer features. It then projects the features back to a lower dimension. This structure, coupled with squeeze-and-excitation (SE) attention that dynamically recalibrates channel-wise feature responses, enhances representational power. The inclusion of skip connections mitigates gradient vanishing, ensuring training stability even in deep architectures. By deploying MBConv in the localization branch of the detection head, we achieve a more favorable trade-off between computational expense and the model’s capacity to precisely regress bounding box coordinates, a critical requirement for distinguishing overlapping pigs.

#### 3.5.2. MBHead

The detection head of YOLOv12 adopts a dual-branch structure, designed for target localization and classification tasks, respectively. After the feature maps output from the neck layer enter the head layer, the localization branch undergoes two standard convolutions (Conv) and one Conv2d convolution, while the classification branch undergoes a combination of two depthwise separable convolutions (DWConv) and standard convolutions (Conv), followed by one Conv2d convolution. Although this design improves detection accuracy, it comes with high computational complexity. Specifically, the total computational load of the YOLOv12-n model is 6.5 GFLOPs, with the detection head accounting for 1.73 GFLOPs, significantly increasing the computational burden.

To address this issue, we propose a lightweight detection head structure called MBHead, whose core innovation is replacing the standard convolution in the localization branch with MBConv. MBConv enhances the model’s ability to capture complex features through an expansion layer that uses 1 × 1 convolution to increase the number of channels (typically with an expansion ratio of 4 or 6). The projection layer then reduces dimensionality using 1 × 1 convolution, retaining critical information. This design is particularly suitable for localization tasks, as the high-dimensional features of the expansion layer help capture details (such as edges and corners), while the projection layer effectively suppresses redundant information. Additionally, the ReLU6 activation function and skip connections (when the input and output channels are consistent) in MBConv enhance gradient flow, mitigate the degradation problem in deep networks, and improve training stability. The depthwise convolution in MBConv operates independently on each channel, implicitly introducing inter-channel independence constraints, which act as a regularization mechanism, reducing the risk of overfitting. This is especially significant in scenarios with limited data. The structure of MBHead and its comparison with the YOLOv12 Head are illustrated in Figure 6.

Experimental results show that the total computational load of the improved model is reduced to 5.3 GFLOPs, with the MBHead detection head accounting for only 0.48 GFLOPs, a reduction of 72.3% compared to the original structure. This improvement significantly enhances the model’s lightweight nature and deployment efficiency while maintaining detection accuracy, making it particularly suitable for real-time object-detection scenarios. Future work will involve further optimizing structural details based on specific task requirements and experimentally validating the optimal configuration.

### 3.6. Improvement of the Loss

In YOLOv12, the bounding box loss employs the CIoU (Complete Intersection over Union) loss function, which is an enhanced variant derived from the traditional IoU (Intersection over Union) loss function. CIoU incorporates an additional penalty term related to the distance between centers and the diagonal length. This enhancement improves the accuracy of CIoU in adapting to bounding boxes of different sizes and geometric configurations. The formulas for calculating IoU and CIoU are as follows:(10)$$\begin{array}{c} IoU=\frac{\;\mathrm{Areaof}\;\mathrm{Intersection}\;}{\;\mathrm{AreaofUnion}\;} \end{array}$$(11)$$\begin{array}{c} CIoU=IoU-\frac{\rho^{2}\left(P,P^{gt}\right)}{c^{2}}-\alpha \cdot m \end{array}$$

Among them, α is the weighting coefficient, *m* measures the similarity of the aspect ratio between the boxes, and $\rho^{2}\left(P,P^{gt}\right)$ represents the distance between the centers of the annotated boxes. The formulas for α and *m* are as follows:(12)$$\begin{array}{c} \alpha =\frac{m}{1-IoU+m} \end{array}$$(13)$$\begin{array}{c} m=\frac{4}{\pi^{2}}\left(\arctan\frac{w^{gt}}{h^{gt}}-\arctan\frac{w}{h}\right) \end{array}$$

CIoU has been widely applied in the field of object detection and has achieved commendable results in numerous applications. However, it still has some limitations in the field of pig behavior recognition. Whether in terms of color or shape, the entire body and limbs of pigs are difficult to delineate, placing higher demands on limb boundary delineation and loss functions. MPDIoU [17] is an efficient boundary regression loss function. The calculation of MPDIoU involves two arbitrary convex shapes, A and B, represented by the coordinates of their top-left and bottom-right corner points. It computes the ratio of the intersection to the union of the two boxes and subtracts the normalized distance between the top-left and bottom-right corner points to obtain MPDIoU. During the training phase, by minimizing the MPDIoU-based loss function $L_{MPDIoU}=1-MPDIoU$, the model predicts each bounding box $B_{prd}$ to be as close as possible to its ground truth box $B_{gt}$.

All factors in existing loss functions (such as non-overlapping regions, center point distance, width–height deviation, etc.) can be determined by the coordinates of the top-left and bottom-right corner points. This means that the proposed $L_{MPDIoU}$ not only considers these factors but also simplifies the computational process. The formula for MPDIoU is as follows:(14)$$\begin{array}{c} MPDIoU=\frac{A\cap B}{A\cup B}-\frac{d_{1}^{2}}{w^{2}+h^{2}}-\frac{d_{2}^{2}}{w^{2}+h^{2}} \end{array}$$
where *A* and *B* are two arbitrary convex shapes, $\left(x_{A1},y_{A1}\right)$ and $\left(x_{A2},y_{A2}\right)$ represent the coordinates of the top-left and bottom-right corner points of *A*, and $\left(x_{B1},y_{B1}\right)$ and $\left(x_{B2},y_{B2}\right)$ represent the coordinates of the top-left and bottom-right corner points of *B*.(15)$$\begin{array}{c} d_{1}^{2}={\left(x_{1}^{B}-x_{1}^{A}\right)}^{2}+{\left(y_{1}^{B}-y_{1}^{A}\right)}^{2} \end{array}$$(16)$$\begin{array}{c} d_{2}^{2}={\left(x_{2}^{B}-x_{2}^{A}\right)}^{2}+{\left(y_{2}^{B}-y_{2}^{A}\right)}^{2} \end{array}$$

The calculation formula for the loss function based on MPDIoU is as follows:(17)$$\begin{array}{c} L_{\mathrm{MPDIoU}\;}=1-MPDIoU \end{array}$$

## 4. Experiments

### 4.1. Datasets

The dataset used in this study was collected through on-site recordings at the Guifei Mountain Farm of Guangxi Yangxiang Co., Ltd. The research team constructed the first pig behavior dataset with fine-grained heat stress annotations, extracting 710 raw images from captured videos by frame sampling. Using the Labelimg tool, pig behaviors were annotated into five categories: standing (stand), sitting (sit), eating (eat), lying down (lying), and heat stress (stress). The annotation of heat stress behavior combines posture features (lying) with physiological characteristics (skin erythema, determined by detecting red channel values exceeding 0.05). Under heat stress conditions, pigs reduce metabolic heat production by decreasing their activity (such as lying down), which is a widely recognized behavioral manifestation of heat stress in academia [1,2,3,4]. Heat stress can cause the dilation of blood vessels in the skin of pigs to dissipate heat, manifesting as erythema on the skin (especially on the abdomen and ears). This characteristic can be quantified through the red channel of images. Through preliminary experiments, we statistically analyzed the distribution of skin red-channel values from 30 pigs confirmed to be under heat stress (rectal temperature ≥ 39.5 °C). A Shapiro–Wilk test confirmed the data followed a normal distribution (W = 0.92, *p* = 0.21). The values (mean ± SD: 0.08 ± 0.02; median: 0.07) had a 95% confidence interval of [0.07, 0.09]. Setting the threshold at 0.05, which is below the lower bound of this interval, ensured coverage of >85% of heat-stressed individuals in our sample, thereby validating its statistical reliability and reducing subjective bias. Dual-feature annotation enriches per-image information beyond posture-only datasets (e.g., Riekert et al., 2020 [8]). Augmentation covers erythema variations under diverse lighting. To enhance the dataset’s diversity and robustness, data augmentation techniques such as flipping and reduced exposure were employed. To address the limitation of single-location data, we supplemented data augmentation strategies that simulate complex farm conditions: random brightness (±30%), contrast (±20%), occlusion (area ≤ 15%), and horizontal flipping (probability = 0.5), which effectively expanded the dataset’s diversity, expanding the dataset to 7100 images. This dataset not only provides high-quality training resources for pig behavior recognition but also offers clear visual feature definitions for heat stress detection, filling the gap in fine-grained animal behavior datasets in agricultural settings and laying a solid foundation for subsequent research. A portion of the dataset is shown in Figure 7.

### 4.2. Experimental Environment and Parameter Setting

Different experimental environments and training parameters may affect the experimental results. To ensure fairness in the experimental comparisons, all experiments in this study were conducted under the same experimental conditions. The details of the experimental environment are listed in Table 1. The training parameters were set as follows: 100 epochs, a batch size of 16, 32 worker threads, an image scaling factor of 0.5, Mosaic data augmentation enabled, a Mixup data augmentation mixing ratio of 0.0, and a copy-paste data augmentation probability of 0.1. The details of the experimental environment are provided in Table 1.

### 4.3. Comparison of Ablation Experiments

To evaluate the effectiveness of the various improvements in the proposed algorithm, this study designed six sets of ablation experiments for comparative analysis. The six sets of experiments include: (1) the original YOLOv12n model; (2) introducing the NAM attention mechanism after the last A2C2f module in the backbone network of YOLOv12n; (3) adding the rectangular self-calibration attention module (RCA) after the second A2C2f module in the Neck network of YOLOv12n; (4) replacing the detection head of YOLOv12n with MBHead; (5) substituting CIoU Loss with MPDIOU Loss; and (6) the complete model incorporating all improvements. The experimental results are shown in Table 2.

The ablation experiment results demonstrate that each improvement positively impacts model performance. Specifically, the introduction of the NAM attention mechanism and the RCA module both increased the precision of heat stress behavior recognition by 0.7%. Although the adoption of the MBHead detection head led to a slight decrease in precision, it significantly reduced the model’s computational complexity and parameter count by 12.5% and 19.0%, respectively. Using the MPDIoU loss function alone improved precision by 0.1%. Introducing both NAM (local feature enhancement) and RCM (spatial reconstruction) simultaneously can lead to redundancy, resulting in increased computational complexity, but it also boosts the mAP gain by 0.9%. Therefore, MBHead can be utilized through MBConv to address this issue, maintaining a lower computational cost compared to the baseline.

Finally, we integrated all the improved modules into the complete model. The introduction of MBHead substantially reduced the parameter count and computational complexity, representing a key improvement for model lightweighting. Combining all enhancements, our model outperformed the original YOLOv12n in terms of detection accuracy, parameter count, and computational complexity. Specifically, the precision of heat stress behavior recognition increased by 0.8%, while the parameter count and computational load decreased by 11.7% and 15.9%, respectively.

To visually demonstrate the detection effects before and after the improvements, we selected several images not involved in training and performed inference using both the original and improved models, with the results shown in Figure 8. Additionally, heatmap detection was conducted, with the results presented in Figure 9. These visualizations further validate the effectiveness of our method. To illustrate the performance of different models in the ablation experiment, we plotted curves for precision and recall, and the results are shown in Figure 10. The results demonstrate the superiority of our proposed model: throughout the entire training process, our PigStressNet model outperforms other comparative models, such as YOLOv12n and YOLOv12n+NAM, in terms of precision and recall. This indicates that our model exhibits superior detection capabilities in detecting heat stress in pigs, enabling it to accurately identify pigs in heat stress conditions while reducing the occurrence of false positives, thus validating the effectiveness of the improvement strategies we adopted.

To verify the biological mechanism of heat stress, we conducted ablation experiments using 300 comparative samples (normal lying/standing erythema/lying erythema). The results are shown in Table 3: only 35% of normal lying was misidentified as heat stress due to posture misjudgment (no erythema detected); only 42% of standing pigs with erythema was misidentified as heat stress due to skin erythema (no posture constraint). The fusion strategy reduced the misjudgment rate of both types to less than 3%, confirming the biological mechanism of heat stress: pigs reduce activity (lying) and dissipate heat through skin vasodilation (erythema) under high temperatures.

To more intuitively evaluate the classification performance of the model in detecting heat stress in pigs, we incorporated a confusion matrix, as illustrated in Figure 11. It can be observed that among the predicted categories, the “stress” category has the highest probability of being correctly predicted, reaching 0.94, indicating that the model excels in identifying stress states. The “lying” category also performs well, with a correct prediction probability of 0.86. However, there is a certain degree of misclassification in the “eat” and “stand” categories. Specifically, the probability of “eat” being misclassified as “stress” is 0.05, and the probability of “stand” being misclassified as “sit” is 0.22. Additionally, the “background” category also exhibits some confusion in prediction, with a probability of 0.15 being misclassified as “lying”. Overall, the model performs well in the “stress” and “lying” categories, but further optimization is needed in the “eat”, “stand”, and “background” categories to reduce misclassification.

### 4.4. Comparison of Detection Performance Between Different Models

To validate the performance advantages of our proposed method in the task of heat stress behavior recognition in pigs, we selected four mainstream object-detection models for comparative experiments with our model, including ShuffleNetV2 [18], EfficientNet [16], GhostNet [19], and MobileNetV3 [20]. The experimental results, as shown in Table 4, clearly demonstrate the comprehensive superiority of our method in terms of detection accuracy, parameter count, and computational complexity.

The results indicate that our method significantly outperforms the other comparative models. Although ShuffleNetV2 excels in model lightweighting, its detection accuracy drops substantially, making it the least accurate model in this experiment. EfficientNet shows some advantages in parameter count but falls short of our model in both precision and computational complexity. GhostNet and MobileNetV3 exhibit mediocre performance in the detection task, lagging behind our model in key metrics such as accuracy, parameter count, and computational complexity. Compared to Dong et al.’s (2022) ST-GCN [12] (lying mAP = 0.89), our fusion method achieves 0.979 stress mAP on identical lying samples by integrating erythema detection, which ST-GCN fundamentally lacks.

## 5. Conclusions

To address the practical challenges in recognizing heat stress behaviors of pigs in current pig farming environments, this paper proposes a lightweight recognition method based on an improved YOLOv12, significantly enhancing the model’s lightweight characteristics while maintaining recognition accuracy. The improvements to the YOLOv12 network include four key aspects:

First, the NAM attention mechanism is introduced after the last A2C2f module in the backbone network of YOLOv12n. Second, the Rectangular Self-Calibration Module (RCM) is added after the second A2C2f module in the Neck network of YOLOv12n, strengthening its feature fusion capability. Additionally, for the detection head, an improved solution based on the Mobile Inverted Bottleneck Block (MBConv) from EfficientNet is proposed, replacing standard convolutions in the localization branch with MBConv to construct a novel detection head, MBHead. Finally, in terms of the loss function, MPDIOU is adopted to replace CIoU, optimizing bounding box regression accuracy.

Experimental results demonstrate that the aforementioned improvements yield significant effects, further validating the method’s comprehensive advantages in lightweight design, detection speed, and accuracy, making it suitable for real-time detection in practical pig farming environments. This work pioneers fusing ’lying posture’ and ’skin erythema’ to define heat stress, resolving single-feature ambiguity. The dual-feature annotated dataset and fusion detection model establish a new paradigm for livestock heat stress monitoring.

However, this study still presents several issues worthy of further exploration. While the proposed MBConv module performs well in MBHead, its application in the backbone or Neck of YOLOv12n yields suboptimal results. Thus, further research is needed to explore variant designs of MBConv and their potential applications in other network architectures.Furthermore, there is still room for improvement in the correlation analysis between this dataset and environmental variables (such as temperature and humidity). In our future research, we will collect more long-term environmental data to deepen this analysis. We firmly believe that this study provides a new paradigm integrating physiological and behavioral features for the field of pig heat stress detection. As the first study focusing on the fusion perspective for pig heat stress detection, it aims to fill the gap in this field. We expect subsequent researchers to build on the fusion framework of this study, conduct in-depth exploration of more complex physiological–behavioral correlation mechanisms, and thus provide a strong impetus for the development of intelligent livestock management.

## Figures and Tables

**Figure 1 sensors-25-05534-f001:**
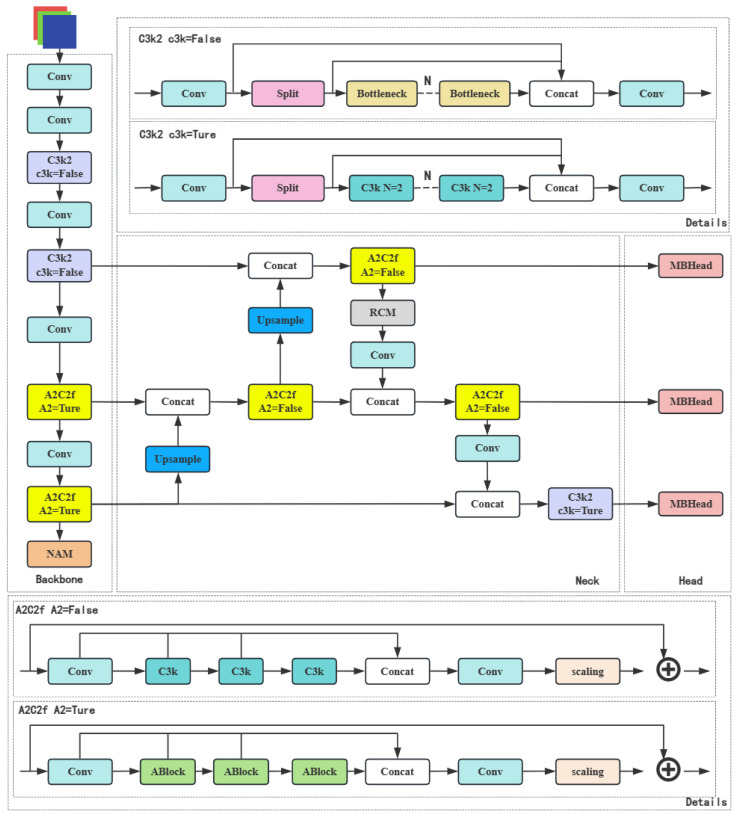
Overall network structure.

**Figure 2 sensors-25-05534-f002:**
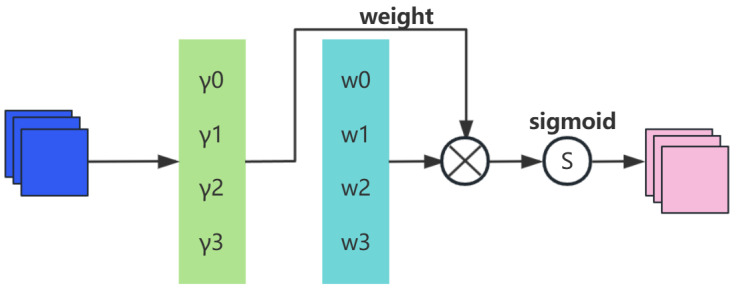
NAM channel attention submodule.

**Figure 3 sensors-25-05534-f003:**
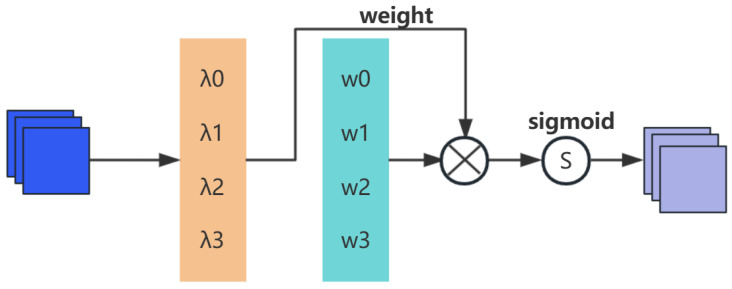
NAM spatial attention submodule.

**Figure 4 sensors-25-05534-f004:**
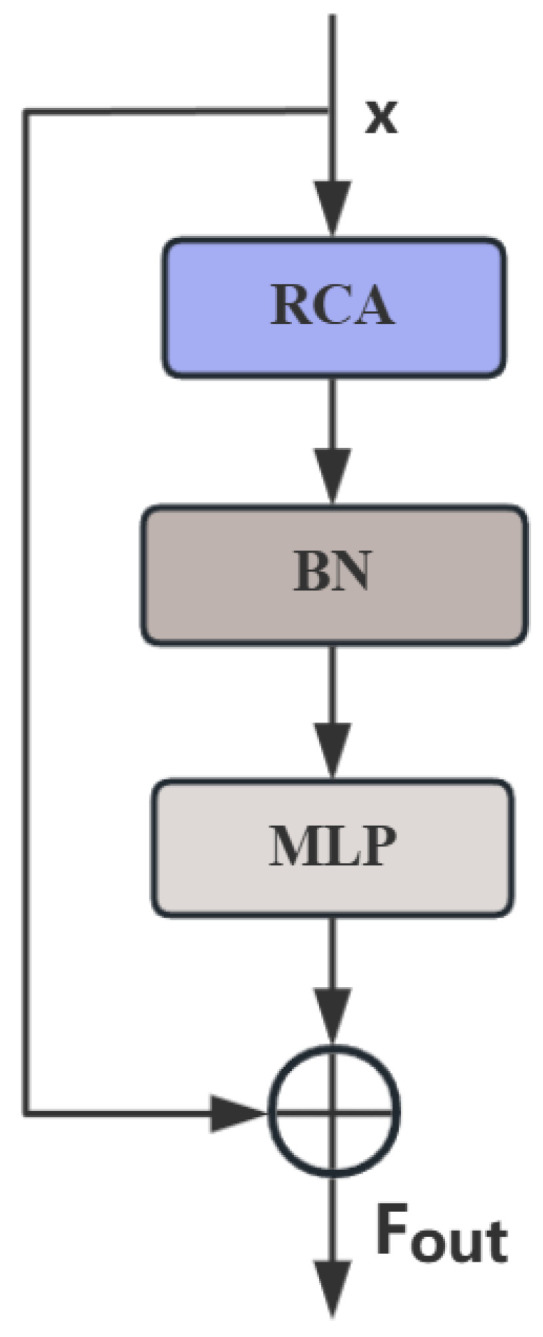
Structure of the RCM Module.

**Figure 5 sensors-25-05534-f005:**
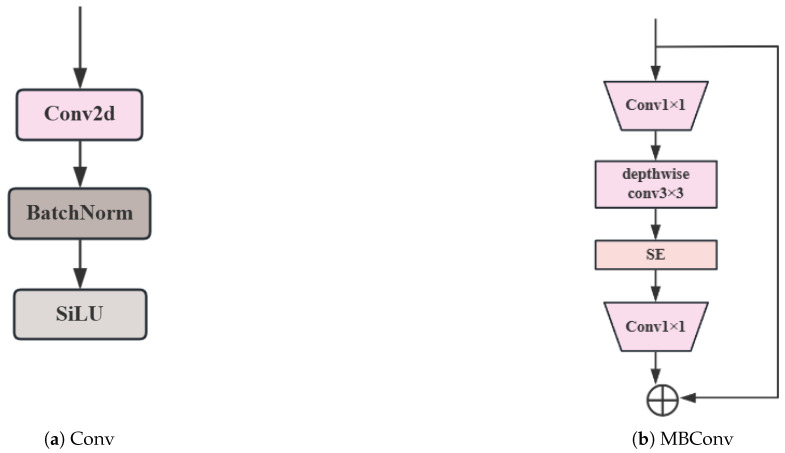
Comparison of MBConv and Conv structures in YOLOv8.

**Figure 6 sensors-25-05534-f006:**
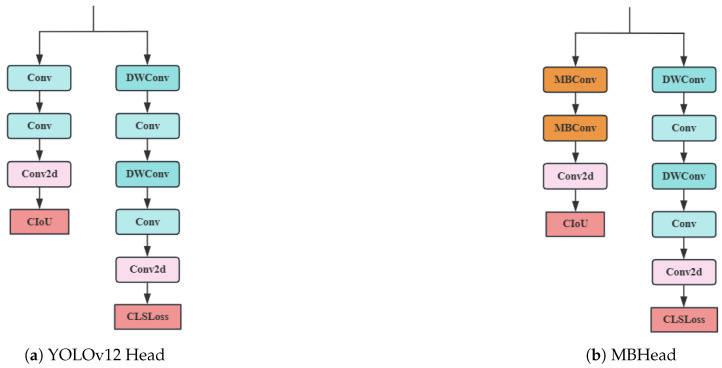
The structure of MBHead and its comparison with the YOLOv12 Head.

**Figure 7 sensors-25-05534-f007:**
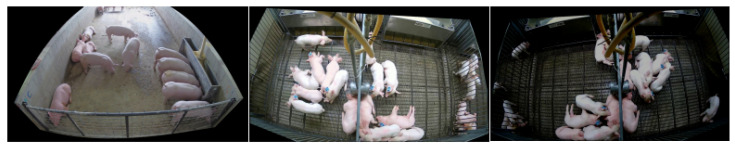
Images from a portion of the dataset.

**Figure 8 sensors-25-05534-f008:**
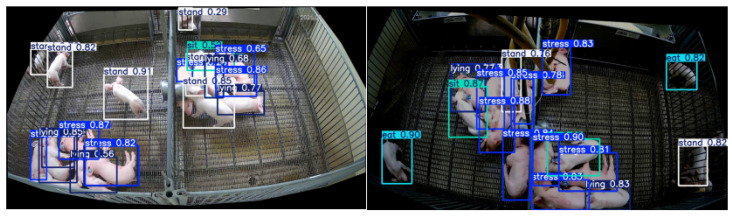
Detection results.

**Figure 9 sensors-25-05534-f009:**
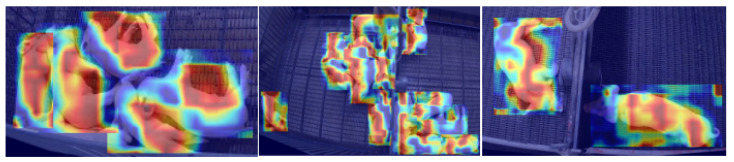
Heatmap detection results.

**Figure 10 sensors-25-05534-f010:**
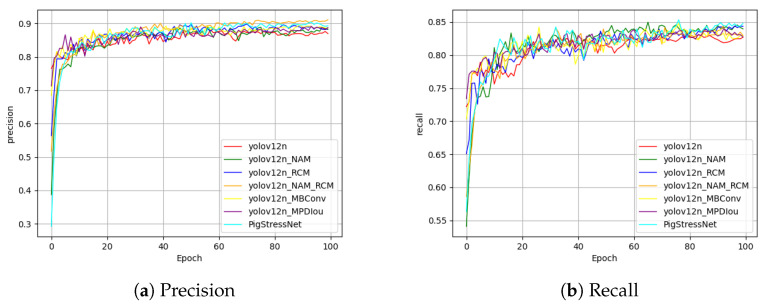
Precision and recall curves.

**Figure 11 sensors-25-05534-f011:**
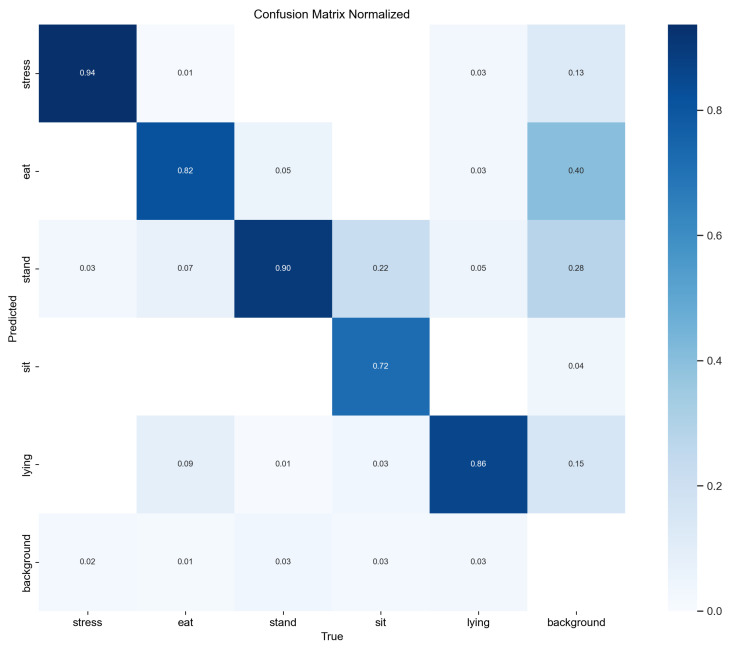
Confusion matrix.

**Table 1 sensors-25-05534-t001:** Experimental environment.

Category	Configuration
CPU	Intel(R)Core(TM)i5-12400F
GPU	NVIDIA GeForce RTX 4060
System environment	Windows 11
Framework	PyTorch 2.6.0
Programming language	Python 3.11
Cuda version	12.4.0

**Table 2 sensors-25-05534-t002:** Comparison of stress ablation results.

Network	mAP					Params (M)	GFLOPs
	Stress	Eat	Stand	Sit	Lying		
YOLOv12n	0.971	0.894	0.870	0.808	0.863	2,557,703	6.3
YOLOv12n+NAM	0.978	0.915	0.876	0.809	0.891	2,558,215	6.3
YOLOv12n+RCM	0.978	0.91	0.895	0.849	0.912	2,576,775	6.6
YOLOv12n+NAM+RCM	0.980	0.917	0.897	0.855	0.905	2,589,375	6.7
YOLOv12n+MBHead	0.969	0.891	0.869	0.806	0.861	2,237,831	5.1
YOLOv12n+MPDIou	0.972	0.899	0.871	0.822	0.864	2,557,703	6.3
PigStressNet	0.979	0.915	0.896	0.853	0.897	2,257,415	5.3

**Table 3 sensors-25-05534-t003:** Feature fusion contribution.

Features	Stress mAP	Misrate (Normal Lying→Stress)	Misrate (Standing Erythema→Stress)
Posture Only	0.75	35%	5%
Erythema Only	0.68	8%	42%
Both	0.979	2%	3%

**Table 4 sensors-25-05534-t004:** Comparative experimental stress result.

Network	mAP					Params (M)	GFLOPs
	Stress	Eat	Stand	Sit	Lying		
ShuffleNetV2 [18]	0.958	0.8	0.799	0.778	0.84	1,711,199	5.0
EfficientNet [16]	0.957	0.842	0.831	0.785	0.822	1,907,451	5.6
GhostNet [19]	0.965	0.85	0.835	0.783	0.889	3,350,159	6.9
MobileNetV3 [20]	0.941	0.873	0.833	0.787	0.85	2,352,241	5.7
PigStressNet	0.979	0.915	0.896	0.853	0.897	2,257,415	5.3

## Data Availability

The raw data supporting the conclusions of this article will be made available by the authors on request.

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
