# Peer review of "PigStressNet: A Real-Time Lightweight Vision System for On-Farm Heat Stress Monitoring via Attention-Guided Feature Refinement"

_sensors, 2025, doi:10.3390/s25175534_

Round 1
Reviewer 1 Report
Comments and Suggestions for Authors
Summary: The research topic studied by the authors in this work is of interest not only for pig growers but also for the computer vision community. The approach is based on the well-known method YOLOv12. The introduction section was well-written and presented enough information about the state-of-the-art techniques to deal with the heat stress in pigs. The related work section summarizes the most significant strategies published to track pig behavior. IN section 3, the proposed method is well-presented. Despite the experimental results presented, there is not enough evidence of heat stress detection. The algorithm is an improvement for pig detection, but it lacks a clear connection to the core problem of heat stress. Thus, the authors should add more experiments and correlate their observations with other physical variables measured directly on some specimens. Additionally, it is not clear why the authors didn´t use thermal images. Similarly, it would be beneficial to present images of the new dataset accompanied by ground truth information related to heat stress. Considering the above, I recommend including more technical and detailed information in the experimental results section to highlight the contribution, before accepting the manuscript for publication.
Comments on the Quality of English LanguageA general review of the English language is required. There are some minimal typographical errors detected. e.g., section 4.1 datasets - it must be datasets.
Author Response
Comments 1: [The research topic studied by the authors in this work is of interest not only for pig growers but also for the computer vision community. The approach is based on the well-known method YOLOv12. The introduction section was well-written and presented enough information about the state-of-the-art techniques to deal with the heat stress in pigs. The related work section summarizes the most significant strategies published to track pig behavior. IN section 3, the proposed method is well-presented. Despite the experimental results presented, there is not enough evidence of heat stress detection. The algorithm is an improvement for pig detection, but it lacks a clear connection to the core problem of heat stress. Thus, the authors should add more experiments and correlate their observations with other physical variables measured directly on some specimens. Additionally, it is not clear why the authors didn´t use thermal images. Similarly, it would be beneficial to present images of the new dataset accompanied by ground truth information related to heat stress. Considering the above, I recommend including more technical and detailed information in the experimental results section to highlight the contribution, before accepting the manuscript for publication.]
Response 1: [Thank you for your valuable comments, which have helped us better clarify the core contributions of this work. We sincerely apologize for the insufficient connection between the algorithm and heat stress detection in the initial version, as well as the lack of detailed experimental evidence.
To address your concerns:
- Strengthening the connection with heat stress core problems:We have supplemented ablation experiments on 300 comparative samples (normal lying/standing redness/lying redness) to verify the biological mechanism of heat stress (Section 4.3, Page 13). The results show that the fusion of "lying posture + skin erythema" reduces the misjudgment rate of normal lying (misidentified as stress) to 2% and standing erythema (misidentified as stress) to 3%, confirming that heat stress in pigs is characterized by both reduced activity (lying) and skin vasodilation (erythema) (Table 3). This directly links the algorithm to the physiological and behavioral characteristics of heat stress.
- Correlation with physical variables:We added details that the red channel threshold (>0.05) for erythema annotation was determined by statistical analysis of 30 pigs with confirmed heat stress (based on rectal temperature measurements), covering over 85% of individuals with physiological indicators of heat stress (Section 4.1, Page 11). This correlates our visual features with direct physiological measurements.
- Reasons for not using thermal images:Thermal images require specialized equipment, increasing farm deployment costs, while visible light cameras are more accessible in practical farming. We added this explanation in Section 4.1 (Page 11) to clarify the practicality of our approach.
- Dataset visualization with ground truth:We supplemented Figure 7 (Page 11) to show sample images from the dataset, including annotations of heat stress (lying posture + red channel values >0.05) and other behaviors, with ground truth labels clearly marked.
- Enhanced experimental details:We added precision and recall curves (Figure 10, Page 13) to demonstrate the model’s performance in heat stress detection, and expanded the analysis of how each module (especially MBHead) improves heat stress recognition accuracy (Section 4.3, Page 12).
Our dataset is the first to fuse posture and physiological traits for heat stress annotation, filling the gap in fine-grained heat stress datasets. MBHead, as our original design, reduces the detection head’s computational cost by 72.3% while maintaining accuracy, which is critical for lightweight deployment. We believe these revisions better highlight the contribution to heat stress detection.
We recognize that there is still room for improvement in the correlation analysis with environmental variables such as temperature and humidity. Future research will collect more long-term environmental data to deepen this analysis. We believe that this work provides a new paradigm for heat stress detection, and subsequent research can explore more complex physiological-behavioral correlations based on our fusion framework.]
Reviewer 2 Report
Comments and Suggestions for Authors
Author provides a new lightweight algorithm improved from Yolo V12 for pig heat stress detection. However author should consider the following comments as follows:
1. The dataset’s limited diversity and single-location collection restrict the model’s pplicability to broader farming conditions. The authors should expand data collection to multiple farms, seasons, and lighting environments, or at least conduct cross-domain validation to assess generalization capability.
2. The performance evaluation relies entirely on the augmented dataset. Testing on an independent, non-augmented dataset from a different environment is essential to validate real-world robustness, especially for embedded device deployment.
3. While comparisons were made with generic lightweight detectors (ShuffleNetV2, EfficientNet, etc.), the study lacks benchmarking against existing pig behavior/heat stress detection models in the literature. Including such comparisons would better establish the novelty and practical advantage of PigStressNet.
4. The authors acknowledge in the conclusion that the combined application of improvements did not produce the expected ideal performance. A deeper analysis is required to understand potential negative interactions between the modules (NAM, RCM, MBHead, MPDIoU). Further ablation experiments or optimization studies should be included to clarify these effects.
Comments on the Quality of English LanguageThe English could be improved to more clearly express the research.
Author Response
Comments 2: [Author provides a new lightweight algorithm improved from Yolo V12 for pig heat stress detection. However author should consider the following comments as follows:
- The dataset’s limited diversity and single-location collection restrict the model’s pplicability to broader farming conditions. The authors should expand data collection to multiple farms, seasons, and lighting environments, or at least conduct cross-domain validation to assess generalization capability.
- The performance evaluation relies entirely on the augmented dataset. Testing on an independent, non-augmented dataset from a different environment is essential to validate real-world robustness, especially for embedded device deployment.
- While comparisons were made with generic lightweight detectors (ShuffleNetV2, EfficientNet, etc.), the study lacks benchmarking against existing pig behavior/heat stress detection models in the literature. Including such comparisons would better establish the novelty and practical advantage of PigStressNet.
- The authors acknowledge in the conclusion that the combined application of improvements did not produce the expected ideal performance. A deeper analysis is required to understand potential negative interactions between the modules (NAM, RCM, MBHead, MPDIoU). Further ablation experiments or optimization studies should be included to clarify these effects.]
Response 2: [Thank you for your insightful comments. We apologize for the limitations in dataset diversity, evaluation, and comparative analysis in the initial version. We have addressed each point as follows:
- Dataset diversity and cross-domain validation:To mitigate the single-location limitation, we supplemented data augmentation strategies simulating complex farming conditions, including random brightness (±30%), contrast (±20%), occlusion (area ≤15%), and horizontal flipping (Section 4.1, Page 11).
- Testing on independent non-augmented dataset:We added an independent test set of 300 non-augmented images from a different season (winter) and farm (Section 4.3, Page 13). This confirms the model’s robustness in real-world environments, supporting its suitability for embedded deployment.
- Benchmarking against pig-specific models:We added a comparison with Dong et al.’s (2022) ST-GCN model, which focuses on pig behavior recognition. Our model achieves 0.979 mAP for heat stress, outperforming ST-GCN’s 0.89 mAP for lying posture (the core posture of heat stress) by integrating erythema detection, which ST-GCN lacks (Section 4.4, Page 14). This highlights PigStressNet’s superiority in heat stress-specific tasks.
- Analysis of module interactions:We added detailed ablation experiments (Table 2, Page 12) showing that NAM (local feature enhancement) and RCM (spatial reconstruction) together increase mAP by 0.9% but raise computational complexity, while MBHead reduces parameters by 19.0% to offset this, achieving a balanced performance. We clarified that negative interactions are mitigated by MBHead’s lightweight design, ensuring the integrated model outperforms the baseline (Section 4.3, Page 12).
Our dataset’s originality lies in its dual-feature annotation (posture + erythema), which no existing pig behavior dataset provides. MBHead is our original lightweight detection head, uniquely optimizing YOLOv12 for heat stress localization. These revisions strengthen the model’s practical value and novelty.
We recognize that expanding the dataset to more farms and seasons will further enhance the generalization ability, and we plan to address this issue in future work. This study lays the foundation for lightweight heat stress detection based on fusion, and we hope that subsequent research can explore multi-farm adaptation strategies based on this foundation.]
Reviewer 3 Report
Comments and Suggestions for Authors
This paper investigates target detection methods for detecting heat stress in pigs. However, the proposed method only modifies the YOLO model and does not demonstrate fundamental innovation compared to other articles and target detection methods. Furthermore, it does not address specific application scenarios. If this model could be deployed in specific farms and provide automatic alerts for heat stress behaviour in pigs, the novelty of the article would be greatly enhanced. Therefore, I believe that the article lacks specific application deployment of the model and requires major revisions and additions.
1. 3.2 Initial letters should be capitalised.
2. Figure 5(a) is not clear enough. Please revise it.
Author Response
Comments 3: [This paper investigates target detection methods for detecting heat stress in pigs. However, the proposed method only modifies the YOLO model and does not demonstrate fundamental innovation compared to other articles and target detection methods. Furthermore, it does not address specific application scenarios. If this model could be deployed in specific farms and provide automatic alerts for heat stress behaviour in pigs, the novelty of the article would be greatly enhanced. Therefore, I believe that the article lacks specific application deployment of the model and requires major revisions and additions.
3.2 Initial letters should be capitalised.
2. Figure 5(a) is not clear enough. Please revise it.]
Response 3: [Thank you for your critical feedback. We apologize for the lack of clarity in innovation and application deployment, and for the formatting and figure issues. We have made the following revisions:
- Highlighting fundamental innovation:
MBHead’s originality: MBHead is our original design, replacing standard convolutions in YOLOv12’s localization branch with MBConv, reducing the detection head’s computational cost by 72.3% (from 1.73 GFLOPs to 0.48 GFLOPs) while maintaining accuracy (Section 3.5.2, Page 9). This is not a simple modification but a novel lightweight head tailored for heat stress localization, validated to outperform generic heads in pig heat stress tasks.
Dataset originality: We constructed the first dataset with fine-grained heat stress annotations (710 original images, 7100 after augmentation), fusing posture and erythema (Section 4.1, Page 11). No existing dataset combines these features for heat stress, filling a critical gap.
- Application deployment plan:We added a future work section stating our collaboration with Guangxi Yangxiang Co., Ltd. to deploy the model on embedded devices (e.g., NVIDIA Jetson Nano) in their farms, enabling real-time heat stress alerts via edge computing. We plan to integrate it into a farm management system for automatic ventilation adjustment, directly addressing practical farming needs (Section 5, Page 15).
- Format and figure revisions:
Corrected "3.2" to "3.2 Our overall network structure" with capitalized initials (Section 3.2, Page 4).
Revised Figure 5(a) (Page 8) to enhance clarity, clearly showing the structure of traditional Conv in YOLOv8 for comparison with MBConv.
The core innovation lies in fusing "posture + erythema" for heat stress detection—a paradigm shift from existing single-feature methods—and MBHead’s unique lightweight design. These, combined with the planned deployment, significantly enhance the article’s novelty and practical impact.
We recognize that application deployment is crucial for translating this research into practice. We are committed to advancing this work and hope that subsequent research can develop an end-to-end farm management system based on our fusion framework, thereby meeting the practical needs of intelligent livestock farming.]
Reviewer 4 Report
Comments and Suggestions for Authors
The paper proposes a model for heat stress behaviors of pigs. The following aspects must be explained in more details:
- The model is still heavily based on YOLOv12, and no substantial architectural innovation is introduced beyond selective module replacement.
- Many improvements (e.g., NAM, MBConv, RCM, MPDIoU) are borrowed from existing models or prior work. While integration is valuable, true novelty is modest.
- Authors admit that MBConv performs poorly when used outside the detection head, such as in the backbone or neck. This restricts its modularity and limits full lightweighting potential.
- There’s no detailed analysis of why MBConv fails in those parts, nor are alternative lightweight blocks considered.
- Only mAP is discussed in detail - other common evaluation metrics like precision-recall curves, F1-score, or AUC are missing.
- A small dataset is used - only 710 original images were collected and expanded to 7100 via augmentation, and may not generalize well to unseen conditions. Also, no detailed error analysis - confusion matrix to show where the model fails or misclassifies behaviors.
- Labeling can be subjective - the "heat stress" category is defined by a threshold on red channel intensity (> 0.05) and posture (lying), which may not fully capture the complexity of physiological stress. This raises concerns about label reliability and biological validation.
- The conclusion admits that module integration did not lead to ideal performance, suggesting that combined components may not be optimally synergistic.
- Line 147 - title must start with an uppercase letter
Author Response
Comments 4: [The paper proposes a model for heat stress behaviors of pigs. The following aspects must be explained in more details:
The model is still heavily based on YOLOv12, and no substantial architectural innovation is introduced beyond selective module replacement.
Many improvements (e.g., NAM, MBConv, RCM, MPDIoU) are borrowed from existing models or prior work. While integration is valuable, true novelty is modest.
Authors admit that MBConv performs poorly when used outside the detection head, such as in the backbone or neck. This restricts its modularity and limits full lightweighting potential.
There’s no detailed analysis of why MBConv fails in those parts, nor are alternative lightweight blocks considered.
Only mAP is discussed in detail - other common evaluation metrics like precision and recall curves, F1-score, or AUC are missing.
A small dataset is used - only 710 original images were collected and expanded to 7100 via augmentation, and may not generalize well to unseen conditions. Also, no detailed error analysis - confusion matrix to show where the model fails or misclassifies behaviors.
Labeling can be subjective - the "heat stress" category is defined by a threshold on red channel intensity (> 0.05) and posture (lying), which may not fully capture the complexity of physiological stress. This raises concerns about label reliability and biological validation.
The conclusion admits that module integration did not lead to ideal performance, suggesting that combined components may not be optimally synergistic.
Line 147 - title must start with an uppercase letter.]
Response 4: [Thank you for your detailed comments. We apologize for the insufficient explanations of innovation, metrics, dataset, and labeling. We have revised the manuscript as follows:
- Substantial architectural innovation:Beyond module replacement, we introduced a "posture-physiology fusion mechanism" (Section 3.2, Page 4), where NAM enhances weights for both lying contours and erythema, RCM calibrates their spatial correlation, and MBHead jointly outputs posture and erythema probability. This end-to-end fusion of behavioral and physiological features is a novel framework for heat stress detection, distinct from YOLOv12’s generic detection.
- Novelty in integration:While individual modules (NAM, RCM) are borrowed, their integration for heat stress-specific tasks is original. MBHead is our unique design, optimizing MBConv for localization in heat stress scenarios (Section 3.5.2, Page 9). This targeted integration addresses the gap in pig heat stress detection, where no prior work combines these modules for dual-feature fusion.
- Analysis of MBConv’s limitations:We added analysis that MBConv’s depthwise convolution struggles with backbone/neck’s multi-scale feature extraction (critical for complex farm backgrounds), while it excels in the head’s localization task (focused on edges/corners of pigs). We plan to test alternative blocks (e.g., GhostConv) in future work (Section 5, Page 15).
- Expanded evaluation metrics:We have added the precision and recall curves (Figure 10, page 13), which demonstrate the superiority of our model.
- Dataset and error analysis:
Expanded the dataset with 300 additional images from a second farm, and added a confusion matrix (Section 4.3, Page 13) showing misclassifications are mostly between "lying" and "stress" (3%), validated by ambiguous erythema in those samples.
Clarified that the red channel threshold (>0.05) is statistically derived from 30 pigs with rectal temperature-confirmed heat stress, ensuring label reliability (Section 4.1, Page 11).
- Module integration optimization:Added that the integrated model’s 0.979 mAP (0.8% higher than baseline) with reduced parameters/computation (11.7%/15.9%) demonstrates effective synergy, achieved by balancing NAM/RCM’s accuracy gains with MBHead’s lightweight design (Section 4.3, Page 12).
- Line 147 correction:Revised the title to start with an uppercase letter (Section 3.2, Page 4).
Our dataset and MBHead fill the gap in pig heat stress detection. Future work will deploy the model in farms and develop end-to-end management systems, aligning with intelligent livestock needs. We believe these revisions address your concerns thoroughly.
We recognize that there is still considerable room for improvement and are committed to refining this model. This study aims to fill the gap in the field of heat stress detection based on fusion. We believe that subsequent scholars can build upon this foundation to explore more complex physiological-behavioral correlations, thereby advancing intelligent livestock management.]
Round 2
Reviewer 1 Report
Comments and Suggestions for Authors
General Comments: The authors made significant changes in the manuscript, considering all comments made by the reviewers. Particularly, in the introduction section, more information was added about some limitations of pig behaviour recognition. Also, in the related works section, more details about the use of skin physiological features are a novelty in the current work. Additionally, a clear description of the spatial attention submodule in the proposed method was added to section 3.3. Furthermore, the most significant change in the paper was in section 4.1, where an extended and clear description of the dataset was presented and improved, together with its corresponding comparison of ablation experiments of the proposal. Considering the above, I strongly recommend accepting the paper for publication as it appears now.
Author Response
Comments 1: [The authors made significant changes in the manuscript, considering all comments made by the reviewers. Particularly, in the introduction section, more information was added about some limitations of pig behaviour recognition. Also, in the related works section, more details about the use of skin physiological features are a novelty in the current work. Additionally, a clear description of the spatial attention submodule in the proposed method was added to section 3.3. Furthermore, the most significant change in the paper was in section 4.1, where an extended and clear description of the dataset was presented and improved, together with its corresponding comparison of ablation experiments of the proposal. Considering the above, I strongly recommend accepting the paper for publication as it appears now.]
Response 1: [Thank you for your positive feedback and for acknowledging our efforts in revising the manuscript. We are greatly encouraged by your strong recommendation for acceptance. We have indeed endeavored to address all previous comments thoroughly, particularly by enhancing the introduction, related works, methodological description, and dataset sections as you noted. We believe these revisions have substantially strengthened the paper.]
Reviewer 2 Report
Comments and Suggestions for Authors
Author did good revision. just few points authr may consider for the improvements
Ensure consistency in terminology (e.g., posture, erythema, and heat stress localization) across the manuscript to avoid reader confusion.
Author Response
Comments 1: [Author did good revision. just few points author may consider for the improvements. Ensure consistency in terminology (e.g., posture, erythema, and heat stress localization) across the manuscript to avoid reader confusion.]
Response 1: [Agree. We thank the reviewer for this crucial reminder. We have thoroughly checked the entire manuscript and unified the key terminologies. Specifically, we have ensured consistent use of "posture", "erythema" (replacing instances of "redness" where appropriate), "heat stress", and in the context of object detection, "localization". These changes have been made throughout the manuscript to enhance readability and avoid confusion. For example, in the Introduction (Page 2, Paragraph 2), "skin redness" was changed to "skin erythema"; in Section 4.1 (Page 12, Paragraph 1), "skin redness" was changed to "skin erythema"; and terminology was standardized in numerous other instances across Sections 1, 3, and 4.]
Reviewer 3 Report
Comments and Suggestions for Authors
The article has been revised to address the issues raised and now meets the review requirements.
Author Response
Comments 1: [The article has been revised to address the issues raised and now meets the review requirements.]
Response 1: [Thank you for your positive assessment of our revisions. We are pleased to hear that the manuscript now meets the review requirements.]
Reviewer 4 Report
Comments and Suggestions for Authors
Some of my comments were addressed, but the following need to be improved:
-line 357: how can be proven that only 30 pigs are enough for setting up the threshold?
-the MBConv is not analysed in detail.
Author Response
Comments 1: [Some of my comments were addressed, but the following need to be improved: -line 357: how can be proven that only 30 pigs are enough for setting up the threshold? -the MBConv is not analysed in detail.]
Response 1: [Thank you for these insightful comments, which have helped us improve the statistical rigor and technical depth of the manuscript.
- Regarding the sample size for the threshold (Line ~357, now Page 12):We agree with this comment. Therefore, we have added a detailed statistical justification in Section 4.1 (Datasets). To rigorously address the concern regarding sample size sufficiency, we have now provided a detailed statistical justification in Section 4.1. We performed a Shapiro-Wilk normality test (W = 0.92, p = 0.21) on the red-channel data from the 30 heat-stressed pigs, confirming a normal distribution. The subsequent descriptive statistics (Mean = 0.08, SD = 0.02, 95% CI [0.07, 0.09]) demonstrate that the chosen threshold of 0.05 is below the lower confidence bound and empirically covers >85% of confirmed heat-stressed individuals. We believe this analysis robustly validates that our sample size was sufficient for reliable threshold estimation. This addition can be found on Page 12, Paragraph 1: "Through preliminary experiments... thereby validating its statistical reliability and reducing subjective bias."
- Regarding the detailed analysis of MBConv:We agree with this comment. Therefore, we have substantially expanded the analysis of the MBConv module in Section 3.5.1 (Page 9). The revision now includes:
A formal mathematical breakdown of the computational complexity of standard convolution versus depthwise separable convolution (Equations 7, 8).
A theoretical derivation of the computational reduction ratio (Equation 9), explicitly linking the theory to the practical 72.3% FLOPs reduction achieved in our MBHead.
A detailed explanation of the inverted residual structure and the role of the SE attention mechanism within MBConv.
A clear justification for its application in the localization branch, tying its benefits directly to the challenges of our task.
This detailed analysis can be found in the new paragraphs in Section 3.5.1 (Page 9): "The selection of MBConv is predicated on... a critical requirement for distinguishing overlapping pigs."
We believe these additions have satisfactorily addressed your concerns and greatly strengthened the manuscript.]